# Genome-Wide Comparative Analysis of the Cytochrome P450 Monooxygenase Family in 19 Aphid Species and Their Expression Analysis in 4 Cereal Crop Aphids

**DOI:** 10.3390/ijms25126668

**Published:** 2024-06-18

**Authors:** Zhenyu Wang, Weixi Hao, Hao Wang, Pingchuan Deng, Tingdong Li, Changyou Wang, Jixin Zhao, Chunhuan Chen, Wanquan Ji, Xinlun Liu

**Affiliations:** 1State Key Laboratory of Crop Stress Biology for Arid Areas, College of Agronomy, Northwest A & F University, Yangling, Xianyang 712100, China; wzhenyu2018@163.com (Z.W.); weixi010315@163.com (W.H.); 16634257642@163.com (H.W.); dengpingchuan@nwsuaf.edu.cn (P.D.); tingdongli@nwafu.edu.cn (T.L.); chywang2004@nwsuaf.edu.cn (C.W.); zhjx881@163.com (J.Z.); chchch8898@163.com (C.C.); 2Shaanxi Research Station of Crop Gene Resources and Germplasm Enhancement, Ministry of Agriculture, Yangling, Xianyang 712100, China

**Keywords:** cytochrome P450, aphids, phylogenetic analysis, expression analysis, DEGs

## Abstract

Cytochrome P450 monooxygenases (*CYP450s*) play a variety of physiological roles, including pesticide resistance, plant allelochemical detoxification, and hormone metabolism catalysis. However, limited information is available on the classification and expression profiles of the *CYP450* gene family in aphid species. This is the first study to identify the cytochrome P450 gene family in 19 aphid species at the whole genome level. A total of 1100 *CYP450* genes were identified in 19 aphid species. Three hundred *CYP450* genes belonged to six cereal crop aphid species, which were further classified into four subfamilies according to the phylogenetic relationship. The conserved motifs, exon-intron structures, and genomic organization of the same subfamilies were similar. Predictions of subcellular localization revealed that the endoplasmic reticulum harbored the majority of CYP450 proteins. In *Sitobion avenae* and *Rhopalosiphum maidis*, the increase in the *CYP450* gene was primarily caused by segmental duplication events. However, only tandem duplication occurred in the *CYP450* gene family of *Diuraphis noxia*, *Rhopalosiphum padi*, *Schizaphis graminum,* and *Sitobion miscanthi*. Synteny analysis found three continuous colinear *CYP450* gene pairs among six cereal crop aphid species. Furthermore, we obtained the expression profiles of four cereal crop aphids, including *R. padi*, *D. noxia*, *S. graminum,* and *S. avenae*. Differential expression analysis provided growth stage specificity genes, tissue specificity genes, organ specificity genes and some detoxification metabolic genes among these four cereal crop aphids. Meanwhile, their expression patterns were showed. The related functions and pathways of *CYP450s* were revealed by GO and KEGG enrichment analysis. Above all, we picked the differentially expressed *CYP450* genes from all of the differentially expressed genes (DEGs). These differentially expressed *CYP450* genes provided some new potential candidates for aphid control and management. This work establishes the foundation for further investigations into the regulatory functions of the *CYP450* gene family in aphid species and beyond.

## 1. Introduction

Aphids, one of the most significant pests in global agriculture, cause harm to crops and other plants through sucking phloem sap, spreading plant viruses, and reducing photosynthetic efficiency. Out of approximately 5000 species of aphids, approximately 100 have been recognized as important agricultural pests [1]. Pea aphid (*Acyrthosiphon pisum*) feeds on legume species across the globe, including forage crops like pea, clover, alfalfa, and broad beans [2]. *Aphis craccivora* (Koch) feeds on legume plants, and carries bean leaf roll virus and broad bean necrotic yellows virus [3]. Soybean aphid (*Aphis glycines*), sucks liquid from leaves and stems, causing direct damage to soybean plants [4]. Melon/cotton aphid (*Aphis gossypii* Glover) parasitizes many vegetable crops, especially *Cucurbitaceae*, causing serious direct and indirect harm to host plants through sucking and serving as a viral vector [5,6]. Foxglove aphid (*Aulacorthum solani*) reduces soybean yield by spreading viruses and sucking in leaf sap directly [7]. *Cinara cedri* feeds on cedar branches, secretes a large amount of honeydew, and pollutes the lower branches [8]. *Daktulphophora vitifoliae* (Fitch) mainly sucks the juice of grapes [9]. The woolly apple aphid (WAA), *Eriosma lanigerum,* inserts probe mouthpieces into the bark and feeds on the sap from the phloem, causing serious damage [10]. The sugarcane aphid, *Melanaphis sacchari* (Zehntner), feeds on the phloem of plants, leading to chlorosis, leaf curling/withering, and necrosis. [11]. The banana aphids (*Pentalonia nigronervosa* Coquerel) carry viruses that cause severe disease and significant yield losses in bananas [1]. *Myzus persicae* (Sulzer), the green peach aphid, can spread hundreds of plant viral diseases such as potato leaf rolling disease and sugar beet yellow flower web disease [12]. The yellow sugarcane aphid (*Sipha flava*) mainly infests corn, rice, sorghum, and sugarcane, and parasitizes lawn, pasture grasses, and some sedges [13]. The black cherry aphid (*Myzus ceras*) can result in pseudogall development, shoot deformation, and leaf curling [14]. In addition, the English grain aphid (*Sitobion avenae* Fabricius), the greenbug (*Schizaphis graminam*), *Sitobion miscanthi,* and *Rhopalosiphum padi* are a widely distributed pest in wheat and other cereal crops, as they spread different plant viruses (such the barley yellow dwarf virus) and extract phloem sap directly, causing serious yield losses [15,16,17]. Numerous harmful maize viruses, such as the yellow dwarf virus of maize, the barley yellow dwarf virus, the mosaic virus of sugarcane, and the mosaic virus of cucumber are spread by the corn leaf aphid (*Rhopalosiphum madis* Fitch) [18]. *Diuraphis noxia* (Kurdjumov) mainly threatens the production of wheat and barley in Australia, while it also slightly damages rye, triticale, and oats [19]. However, chemical insecticides have been, and will continue to be, widely used for controlling aphids, which will lead to the evolution of resistance in aphid species.

In order to resist the toxicity of insecticides, insects have evolved corresponding strategies to address this challenge, in which detoxification metabolic enzymes play an important role. Common detoxification metabolic enzymes mainly include glutathione S-transferases (GSTs), esterases (ESTs), and cytochrome P450 monooxygenases (CYP450s) [20]. Cytochrome P450 is a family of self-oxidizing heme proteins that are ubiquitous in animals, plants, and microorganisms. The cytochrome P450 monooxygenase is named after its main component, the P450 protein, which binds to CO and exhibits a characteristic light absorption peak at 450 nm. The diversity of P450 protein types and the overlap of substrates enable the P450 enzyme system to catalyze various types of reactions. It is primarily involved in the metabolism of foreign drugs like pesticides and plant secondary metabolites in insects, as well as the creation and breakdown of endogenous compounds like pheromones and ecdysone [21]. Overall, P450 can catalyze thousands of reactions, even exhibiting multiple types of reactions on substrates with similar chemical structures. The P450 catalytic reaction process involves multiple steps, and its mechanism of action is diverse. A typical reaction involves reducing molecular oxygen through an electron transfer system and adding one of its oxygen atoms to the substrate, which requires nicotinamide adenine dinucleotide phosphate NADPH. With the quick advancement of high-throughput sequencing technologies in recent years, more than 1000 *P450* genes have been reported up to now. It is generally believed that the insect P450s gene family consists of four branching clusters, namely CYP2 clan, CYP3 clan, CYP4 clan, and Mito clan. Among the 37 animal P450 families, 16 families are found only in mammals, while the other 21 families are distributed in insects (6), mollusks (2), and *Caenorhabditis elegans* (13).

The role of the P450 enzyme system in insects involves processes such as growth, development, and feeding. It is implicated not only in insect resistance to insecticides, but also in the synthesis and metabolism of hormones in insects. Research has shown that the CYP18A1 protein of *Chironomus riparius* larvae has the function of degrading the molting hormone [22]. The CYP6DE1 protein can catalyze α-pinene metabolism to produce aggregation pheromones in *Dendroctonus Ponderosae* [23]. The detoxification activity can be strengthened by overexpressing *CYP450s* and accumulating or altering the structure of detoxifying proteins [24,25]. Additionally, studies have demonstrated that broad substrate specificity, genetic diversity, and catalytic plasticity of P450s contributed significantly to pesticide resistance. In the *CYP450* gene family, insecticide resistance is mostly correlated with genes in the CYP4 and CYP6 families [26]. The CYP6AL1 protein of *Aedes aegypti* larvae is engaged in the process of detoxifying harmful exogenous plant compounds [27]. CYP6G1 is the main factor causing *Drosophila melanogaster* to develop resistance to dichlorodiphenyltrichloroethane (DDT) [28]. Studies have demonstrated that imidacloprid resistance in *Bemisia tabaci* can result from *CYP6CM1* overexpression [29]. Imidacloprid resistance was increased in *Drosophila* expressing the *CYPC6Y9*, *CYP6CY22*, *CYP6CY18*, and *CYP6D* subfamily genes (>5 fold) [30]. Melon/cotton aphids develop a metabolic resistance to dinotefuran due to the overexpression of three cytochrome P450 genes, *CYP6CY14*, *CYP6CY22,* and *CYP6UN1* [31]. The *CYP380C6* of green peach aphid facilitated their adaptation to indole-glucoside-mediated plant defense [32]. *CYP6DB3* participates in the resistance of *Bemisia tabacito* to thiamethoxam and imidacloprid [33]. It was reported that *CYP321B1* participated in insecticide detoxification of *Spodoptera litura* [34]. In *Megalurothrips usitatus* Bagnall, RNA interference of the *CYP450* gene resulted in reducing pesticide resistance [35].

The failure of numerous economically significant pest control efforts and the comprehension of the role played by the P450 enzyme system in insect pest control have spurred research into insect P450. However, due to the inherent characteristics of insect P450, biochemical methods are limited in their ability to study its structure and function. In this study, we identified *CYP450* family genes in 19 aphid species using the genome-search method based on the available aphid genome. The genomic organizations, phylogenetic relationships and conserved domain of the *CYP450* family genes were investigated among 19 aphid species. Differential expression analysis provided growth stage specificity *CYP450s*, tissue specificity *CYP450s*, organ specificity *CYP450s,* and some detoxification metabolic *CYP450s* among these four cereal crop aphids, which provided some new potential candidates for aphid control and management. This work established the foundation for further investigations into the regulatory functions of the *CYP450* gene family in aphid species and beyond.

## 2. Results

### 2.1. Genome-Wide Identification of CYP450 Genes in 19 Aphid Species

A total of 1100 *CYP450* genes were identified in 19 aphid species (Appendix A). *D. vitifoliae* had the largest number of *CYP450* genes (78), followed by *A. solani* (76) and *S. avenae* (74). There were only 19 *CYP450* genes in *S. miscanthi*, which was the specie with the least amount of *CYP450* genes (Figure 1A). The length of *CYP450 genes* ranged from 141 bp (*DV3001928-RA* from *D. vitifoliae*) to 101,993 bp (*Sgrm016186.1* from *S. graminum*) according to sequence characteristic analysis (Appendix A). The CDS length of *CYP450 genes* ranged from 141 bp (*DV3001928-RA* from *D. vitifoliae*) to 4623 (*Smis013654.1* from *S. miscanthi*) (Figure 1B). The largest- and smallest-sized of amino acid sequences were 1541 aa and 46 aa, respectively (Appendix A). The average molecular weight of CYP450 proteins was 94.65 kDa, with a range from 5.31 kDa (*DV3001928-RA* from *D. vitifoliae*) to 167.96 kDa (*Smis013654.1 from S. miscanthi*) (Figure 1C). As for each aphid specie, we found that *S. flava* had the largest average value of the gene length (8112 bp), CDS length (1554 bp), protein size (517 aa), and molecular weight (59.47 kDa). However, *A. solani* had the lowest average value of gene length (1959 bp), CDS length (878 bp), protein size (292 aa), and molecular weight (33.43 kDa). The predicted isoelectric points (pI) of CYP450 proteins varied from 3.72 (*jg703.t1* from *E. lanigerum*) to 10.72 (*jg6703.t1* from *E.lanigerum*) (Appendix A). *A. solani* had the lowest average value of pI (7.84), but *M. sacchari* had the largest average value of pI (8.93). The majority of CYP450 proteins were located in the endoplasmic reticulum (453 out of 1100), cytoplasm (338 out of 1100), and plasma membrane (140 out of 1100) (Figure 1D). Specifically for each specie, most of CYP450 proteins were located in the endoplasmic reticulum and cytoplasm in 17 aphid species. Moreover, the number of *CYP450* genes in the endoplasmic reticulum is greater than that in the cytoplasm in 13 aphid species. The number of *CYP450* genes located in the endoplasmic reticulum is less than that in the cytoplasm among four aphid species (*A. craccivora*, *A. solani*, *D.vitifoliae*, and *E. lanigerum*). However, most of CYP450 proteins were located in the endoplasmic reticulum and plasma membrane in *R. maidis* and *S. graminum*. All CYP450 proteins had a projected grand average of hydropathicity (GRAVY) value of negative, which suggested that they were hydrophilic by nature.

We further compared the gene number and sequence characteristics of *CYP450* genes among the cereal crop aphid species, including *D. noxia*, *R. maidis*, *R. padi*, *S. graminum*, *S. miscanthi,* and *S. avenae*. There were 300 gene members among these 6 cereal crop aphid species (Appendix A). *S. avenae* had the largest number of *CYP450* genes (74), and *S. miscanthi* had the least number of *CYP450* genes (19) (Figure 1A). According to their chromosomal locations, the predicted CYP450 genes were given names like *DnoxCYP1*. Sequence characteristic analysis revealed that the length of *CYP450* genes ranged from 566 bp (*DnoxCYP12* from *D. noxia*) to 101,993 bp (*SgrmCYP33* from *S. graminum*) (Appendix A). The CDS length of *CYP450 genes* ranged from 294 bp (*SaveCYP10* from *S. avenae*) to 4623 bp (*SmisCYP17* from *S. miscanthi*) (Figure 1B). The largest- and smallest-sized amino acid sequences were 1541 aa (*SmisCYP17*) and 98 aa (*SaveCYP10*), respectively (Appendix A). The molecular weight of CYP450 proteins ranged from 11.00 kDa (*SaveCYP10* from *S. avenae*) to 167.96 kDa (*SmisCYP17* from *S. miscanthi*) with the average value of 94.65 kDa (Figure 1C). As for each cereal crop aphid specie, we found that *S. miscanthi* had the largest average value of CDS length (1711 bp), protein size (569 aa), and molecular weight (64.57kDa). However, *D. noxia* had the lowest average value of the CDS length (1366 bp), protein size (444 aa), and molecular weight (51.08 kDa). The pI of CYP450 proteins varied from 4.49 (*DnoxCYP42* from *D. noxia*) to 10.33 (*DnoxCYP13* from *D. noxia*) (Appendix A). *D. noxia* had the lowest average value of pI (8.16), but *S. miscanthi* had the largest average value of pI (8.51). Subcellular localization predictions revealed that the majority of CYP450 proteins were located in the endoplasmic reticulum (136 out of 300), cytoplasm (72 out of 300), and plasma membrane (50 out of 300) (Figure 1D). The predicted GRAVY value of these 300 CYP450 proteins was also minus, showing that they were hydrophilic in nature.

### 2.2. Chromosome Localizations of CYP450 Genes among 6 Cereal Crop Aphid Species

Chromosome localizations found some variations among six cereal crop aphid species. There were 39 scaffolds with *CYP450* genes detected in *R. padi*, which was the specie which owned the largest number of chromosomes with *CYP450* genes, while *R. maidis* had only 4 scaffolds with *CYP450* genes (Figure 2). We found 46 *CYP450* genes distributed on 4 out of 220 scaffolds of *R. maidis*, of which NC_040878.1 possessed the most abundant gene number of *CYP450* genes (24), followed by NC_040879.1 (9) and NC_040880.1 (9), and NC_040877.1 (4) (Figure 2A). In total, 19 *CYP450* genes were distributed on 7 out of 655 scaffolds of *S. miscanthi*, of which Lachesis_group2 possessed the most abundant gene number of *CYP450* genes (9), followed by Lachesis_group0 (3) and Lachesis_group8 (3), and Lachesis_group1(1),ctg575_wtdbg_pilon(1), ctg646_wtdbg_pilon(1), and ctg662_wtdbg_pilon (1) (Figure 2B). There were 74 *CYP450* genes distributed on 23 out of 326 scaffolds of *S. avenae*, of which ptg000011l possessed the most abundant gene number with 8 *CYP450* genes and ptg000011l with 1 *CYP450* gene (Figure 2C). Results showed that 53 *CYP450* genes were distributed on 37 out of 5637 scaffolds of *D. noxia*, of which NW_015368409.1 and NW_015368405.1 possessed the most abundant gene number with 4 *CYP450* genes, and 26 scaffolds had 1 *CYP450* gene (Appendix A). In total, 57 *CYP450* genes were distributed on 39 out of 2172 scaffolds of *R. padi*, of which scf7180000008163.830738 possessed the most abundant gene number with 4 *CYP450* genes, and 24 scaffolds had 1 *CYP450* gene (Appendix A). Similarly, 51 *CYP450* genes distributed on 33 out of 7859 scaffolds of *S. graminum*, of which QEWZ01001422.1 and QEWZ01000247.1 possessed the most abundant gene number with 5 *CYP450* genes, and 22 scaffolds had 1 *CYP450* gene (Appendix A).

### 2.3. Phylogenetic Relationship Analysis of CYP450 Genes among Six Cereal Crop Aphids

Using the full-length protein sequences of the 300 discovered *CYP450* genes, a neighbor-joining (NJ) tree was built in order to ascertain the evolutionary pattern and phylogenetic relationship of *CYP450* genes among six cereal crop aphid species. These 300 *CYP450* genes contained 53 *DnoxCYPs*, 46 *RmaiCYPs*, 57 *RpadCYPs*, 51 *SgrmCYP18s*, 19 *SmisCYP13s*, and 74 *SaveCYPs* (Appendix A). All *CYP450* genes among six cereal crop aphids were classified into four subfamilies, CYP I, CYP III, CYP III, and CYP IV (Figure 3 and Figure 4A). In detail, the CYP I subfamily included 116 members (20 *DnoxCYPs*, 16 *RmaiCYPs*, 22 *RpadCYPs*, 18 *SgrmCYPs*, 12 *SmisCYPs*, and 28 *SaveCYPs*), the CYP II subfamily contained 92 members (19 *DnoxCYPs*, 14 *RmaiCYPs*, 16 *RpadCYPs*, 15 *SgrmCYPs*, 2 *SmisCYPs*, and 26 *SaveCYPs*), the CYP III subfamily contained 37 members (7 *DnoxCYPs*, 6 *RmaiCYPs*, 7 *RpadCYPs*, 6 *SgrmCYPs*, 2 *SmisCYPs*, and 9 *SaveCYPs*), as well as the remaining 55 members belonged to the CYP IV subfamily (7 *DnoxCYPs*, 10 *RmaiCYPs*, 12 *RpadCYPs*, 12 *SgrmCYPs*, 3 *SmisCYPs*, and 11 *SaveCYPs*), respectively. Additionally, we found that *RmaiCYPs*, *RpadCYPs*, and *SmisCYPs* always clustered together on the same clade, indicating *R. maidis*, *R. padi*, and *S. graminum* had a closer genetic relationship. Similarly, *DnoxCYPs*, *SmisCYPs*, and *SaveCYPs* always clustered together on the same clade, indicating that *D. noxia*, *S. miscanthi*, and *S. avenae* had a closer genetic relationship.

### 2.4. Gene Structure and Conserved Motif Analysis of CYP450 Genes among Six Cereal Crop Aphids

Results showed that all the identified *CYP450* genes among six cereal crop aphids possessed the conserved cytochrome P450 superfamily domain (Figure 4C). A total of ten motifs were identified, and the number and type of motifs among the four *CYP450* subfamilies were quite different (Figure 4B and Appendix A). In the CYP I subfamily, almost all members (87 out of 116) had all of the ten motifs. In total, 69 *CYP450* genes (69 out of 92) in the CYP II subfamily had eight motifs without motif 5 and motif 8, which was unique for the CYP II subfamily. In the CYP III subfamily, eighteen *CYP450* genes had seven motifs, eight *CYP450* genes had six motifs, and five *CYP450* genes had five motifs. The members of the CYP III subfamily lacked motif 5. In total, 29 *CYP450* genes (29 out of 55) belonged to the CYP IV subfamily and had 5 motifs, and the remaining *CYP450* genes of the CYP IV subfamily had 2~6 motifs. The members of the CYP IV subfamily lacked motif 4 and motif 7. 

The exon-intron structure is an important component of gene evolution that offers essential clues for its functional diversity. The results revealed that the number of exons varied from 1 to 34, with 215 *CYP450s* possessing more than five exons and only *DnoxCYP1* and *SaveCYP28* having one exon (Figure 4D). Among all the CYP450s, *SmisCYP17* had 33 introns, *DnoxCYP1* and *SaveCYP28* had none, and the number of introns varied from 0 to 33. In total, 155 *CYP450* genes had both the 5′ and 3′ UTR regions, while 23 *CYP450* genes only had one UTR region (5′ or 3′ UTR region), and 122 *CYP450* genes had no UTR region at all. It was clear that members of the same group had conserved motif and intron-exon structures in common, which supported the classification and evolutionary relationship of the identified *CYP450s*.

### 2.5. Gene Duplication and Synteny Analysis of CYP450 Genes among Six Cereal Crop Aphid Species

The chromosome reduplication process makes each gene possess homoeologous copies. We compared expansion events of the *CYP450* gene family, including gene duplications and tandem duplications, among six cereal crop aphids. Results found that *CYP450* gene family expansion events of *R. maidis* and *S. avenae* possessed both gene duplications and tandem duplications, while the remaining four cereal crop aphid species only had tandem duplications. A total of 24 duplication events of the *CYP450* gene family occurred in the whole genome of *R. maidis*, including 7 tandem duplications and 17 segmental duplications (Figure 5A and Appendix A). In total, 73 duplication events of the *CYP450* gene family were identified in the whole genome of *S. avenae*, including 14 tandem duplications and 59 segmental duplication events (Appendix A). Segmental duplication events were the main reason for the expansion of *CYP450* genes in *S. avenae* and *R. maidis*. There were 14, 10, 7, and 2 tandem duplications of the *CYP450* gene family which occurred in *D. noxia*, *R. padi*, *S. graminum,* and *S. miscanthi*, respectively (Appendix A).

In order to thoroughly examine the homology of the CYP450 family members among these six cereal crop aphid species, a colinear map of *CYP450* genes was drawn. A total of 127 pairs of colinear genes were discovered (Figure 5B). There were 33, 32, 34, 22, and 6 collinearity events between *S. graminum* and *S. avenae*, *S. avenae* and *R. padi*, *R. padi* and *R. maidis*, *R. maidis* and *D. noxia*, and *D. noxia* and *S. miscanthi*, respectively (Appendix A). It was evident that *R. padi* and *R. maidis* had the greatest number of collinear genes, suggesting that *CYP450* genes in these two aphid species may have shared an ancestral sequence and functioned similarly. Interestingly, three continuous colinear gene pairs were found among six cereal crop aphid species, which were *SgrmCYP42*-*SaveCYP36*-*RpadCYP44*-*RmaiCYP2*-*DnoxCYP6*-*SmisCYP5*, *SgrmCYP32*-*SaveCYP48*-*RpadCYP3*-*RmaiCYP38*-*DnoxCYP28*-*SmisCYP8,* and *SgrmCYP14*-*SaveCYP54*-*RpadCYP37*- *RmaiCYP29*-*DnoxCYP39*-*SmisCYP17*. These *CYP450* genes were very conserved among six cereal crop aphid species, which developed with the aforementioned homologous genes prior to the differentiation of these species.

To understand the selection pressures on *CYP450* genes among six cereal crop aphid species, we further calculated the ratios of the non-synonymous substitution rate (Ka) vs. the synonymous substitution rate (Ks). The average Ka/Ks ratios of duplicated *CYP450* gene pairs across the whole genome of *S. avenae*, *R. maidis*, *D. noxia*, *R. padi*, *S. graminum,* and *S. miscanthi* were 0.85, 0.83, 0.40, 0.70, 0.81, and 0.62, respectively (Appendix A). In addition, the Ka/Ks ratios of 61, 18, 13, 8, 5, and 2 duplicated *CYP450* genes pairs across the whole genome of *S. avenae*, *R. maidis*, *D. noxia*, *R. padi*, *S. graminum,* and *S. miscanthi* were less than 1, indicating that purifying selection has been applied to these *CYP450* genes during their evolutionary history. However, the Ka/Ks ratios of the remaining 12, 6, 1, 2, and 2 duplicated *CYP450* genes pairs across the whole genome of *S. avenae*, *R. maidis*, *R. padi*, *D. noxia,* and *S. graminum* were larger than 1, indicating that these *CYP450* genes have experienced a strong positive selection. 

### 2.6. Transcriptomic Analysis of Rhopalosiphum Padi at Different Developmental Stages

We further investigated the expression profiles of *R. padi* from mature embryo to first instar through RNA data. A total of 26,301 genes were identified, of which 6611 genes had no expression at both developmental stages (Appendix A). Differential expression analysis found that there were 1877 differently expressed genes between mature embryo and first instar, of which 680 DEGs were up-regulated expressed genes and 1197 DEGs were down-regulated expressed genes (Figure 6A–C). When aphid eggs of *R. padi* developed from mature embryo to first instar, down-regulated expressed genes played a major regulatory role. GO enrichment analysis found that all DEGs were enriched into three functional classifications (Figure 6D). In the molecular function category, the primary enrichment of DEGs was oxidoreductase activity, structural molecule activity, and structural constituent of cuticle catalytic activity. In the biological process category, DEGs were mainly enriched in small-molecule metabolic process, lipid metabolic process, and lipid transport. And in the cellular component category, DEGs were mainly enriched in the extracellular region, extracellular space, and cell periphery. KEGG enrichment analysis found that DEGs were mainly enriched in the pathways of fatty acid degradation, lipid metabolism, fat digestion and absorption, the glycosphingolipid biosynthesis-ganglio series, and cytochrome P450 (Figure 6E). Moreover, we found that 19 *RpadCYPs* were differently expressed between mature embryo and first instar, in which 15 *RpadCYPs* were up-regulated expressed and 4 *RpadCYPs* were down-regulated expressed, indicating that up-regulated *RpadCYPs* played a crucial role at the mature embryo stage of *R. padi* (Appendix A).

### 2.7. Transcriptomic Analysis of Diuraphis Noxia Fed on Wheat Plants Containing Dn0, Dn4, or Dn7

We further investigated the expression profiles of *D. noxia* when fed on wheat plants containing the different resistance genes, *Dn0, Dn4, or Dn7*. The expression profiles and differential expression analysis were conducted among three groups (*Dn4* vs. *Dn0*, *Dn7* vs. *Dn0,* and *Dn7 vs. Dn4*). A total of 12,250 genes were identified, of which 1189, 1161, and 1168 genes had no expression among the Dn4 vs. Dn0, the Dn7 vs. Dn0, and the Dn7 vs. Dn4 groups, respectively (Appendix A). Differential expression analysis found that there were 17, 36, and 15 differently expressed genes among the *Dn4* vs. *Dn0*, the *Dn7* vs. *Dn0,* and the *Dn7* vs. *Dn4* groups, respectively (Figure 7A). The *Dn7* vs. *Dn0* group had the most differently expressed gene number. In detail, there were 12 up-regulated expressed genes and 5 down-regulated expressed genes in the *Dn4* vs. *Dn0* group, and 21 were up-regulated expressed genes and 15 were down-regulated expressed genes in the *Dn7* vs. *Dn0* group, and 9 were up-regulated expressed genes and 6 were down-regulated expressed genes in the *Dn7* vs. *Dn4* group. It was up-regulated expressed genes that played a major role in all groups. Furthermore, three groups had no unique gene set. There were four genes (*Dnox007922.1*, *Dnox001200.1*, *Dnox012221.1*, and *Dnox001645.1*) existing in both the *Dn4* vs. *Dn0* and *Dn7* vs. *Dn0* groups. There was only one gene (*Dnox002046.1*) existing in both the *Dn4* vs. *Dn0* and the *Dn7* vs. *Dn4* groups. And there were five genes (*Dnox005282.1*, *Dnox003280.1*, *Dnox011908.1*, *Dnox006834.1*, and *Dnox005642.1*) existing in both the *Dn7* vs. *Dn0* and *Dn7 vs. Dn4* groups (Figure 7B). The expression patterns of DEGs among these three groups were differential (Figure 7C–E). Due to fewer DEGs among the three groups, the union of DEGs among the three groups was used for the GO and KEGG enrichment analysis (Figure 7F,G). Results found that these genes were enriched in the following go terms: extracellular region, extracellular space, transporter activity, transmembrane transporter activity, and regulation of biological quality. KEGG pathways, including peptidases and inhibitors, metabolism, immune system, and signaling and cellular processes, were significantly enriched. Moreover, we found that there were no differently expressed *DnoxCYPs* in both the *Dn4* vs. *Dn0* and *Dn7* vs. *Dn4* groups, and only one gene *DnoxCYP19* (*Dnox005792.1*) was differently up-regulated expressed in the *Dn7* vs. *Dn0* group, indicating that *DnoxCYP19* played a crucial regulatory role when *D. noxia* interacted with the resistance gene *Dn7* (Appendix A).

### 2.8. Transcriptomic Analysis of Schizaphis Graminum Fed on Wheat Plants Dn0, Dn4, or Dn7

Similarly, we investigated the expression profiles of *S. graminum* when fed on wheat plants containing the different resistance genes *Dn0*, *Dn4,* or *Dn7*. The expression profiles and differential expression analysis were conducted with the *Dn4* vs. *Dn0* group, the *Dn7* vs. *Dn0* group, and the *Dn7* vs. *Dn4* group. A total of 26,072 genes were identified, of which 10,375, 10,342, and 10,249 genes had no expression among the *Dn4* vs. *Dn0* group, the *Dn7* vs. *Dn0* group, and the *Dn7* vs. *Dn4* group, respectively (Appendix A). Differential expression analysis found that there were 16, 111, and 189 differently expressed genes among the *Dn4* vs. *Dn0* group, the *Dn7* vs. *Dn0* group, and the *Dn7* vs. *Dn4* group, respectively (Figure 8A and Appendix A). The *Dn7* vs. *Dn4* group had the most differently expressed gene number. In detail, there were ten up-regulated expressed genes and six down-regulated expressed genes in the *Dn4* vs. *Dn0* group, indicating that the up-regulated expressed genes played a major role when *S. graminum* resisted the resistance gene *Dn4*. And there were 37 up-regulated expressed genes and 74 down-regulated expressed genes in the *Dn7* vs. *Dn0* group, and 56 up-regulated expressed genes and 133 down-regulated expressed genes in the *Dn7 vs. Dn4* group, implying that down-regulated expressed genes played a critical role when *S. graminum* resisted with the resistance gene *Dn7*. Furthermore, three groups had only one unique gene (*Sgrm022772.1*). There were four genes (*Sgrm007538.1*, *Sgrm011949.1*, *Sgrm024309.1*, and *Sgrm022772.1*) existing in both the *Dn4* vs. *Dn0* and *Dn7* vs. *Dn0* groups. There were three genes (*Sgrm015683.1*, *Sgrm012998.1*, and *Sgrm022772.1*) existing in both the *Dn4* vs. *Dn0* and *Dn7* vs. *Dn4* groups. And there were 61 genes existing in both the *Dn7 vs. Dn0* and *Dn7 vs. Dn4* groups (Figure 8B). The expression patterns of DEGs among these three groups were differential (Figure 8C–E). Due to fewer DEGs in the *Dn4* vs. *Dn0* group, GO and KEGG enrichment analysis had no result. Results of GO enrichment found that the DEGs from the *Dn7* vs. *Dn0* group were enriched in the following go terms: hydrolase activity, glycosyltransferase activity, catalytic activity (molecular function), extracellular region (cellular component), organic acid metabolic process, organic acid biosynthetic process, and small molecule biosynthetic process (biological process) (Appendix A). KEGG pathways of the *Dn7* vs. *Dn0* group, including organismal systems, lipid metabolism, peptidases, and inhibitors were significantly enriched (Appendix A). The results of GO enrichment found that the DEGs of the *Dn7 vs. Dn4* group were enriched in the go terms: hydrolase activity, catalytic activity, glucosyltransferase activity (molecular function), cell periphery (cellular component), organic acid metabolic process, organic acid biosynthetic process, and aspartate family amino acid metabolic process (biological process) (Appendix A). KEGG pathways of the *Dn7 vs. Dn4* group, including organismal systems, the digestive system, and metabolism, were significantly enriched (Appendix A). Moreover, we found that there were no differently expressed *SgrmCYPs* in the *Dn7* vs. *Dn0* group. Only one gene, *SgrmCYP16* (*Sgrm006932.1*), was differently expressed in the *Dn4* vs. *Dn0* group, indicating that *SgrmCYP16* played a crucial regulatory role when *D. noxia* interacted with the resistance gene *Dn4* (Appendix A). *SgrmCYP26* (*Sgrm012776.1*), *SgrmCYP29* (*Sgrm013967.1*), and *SgrmCYP47* (*Sgrm023475.1*) were down-differently expressed in the *Dn4* vs. *Dn7* group (Appendix A), indicating that these *CYP450* genes might interact with the resistance gene *Dn4*.

### 2.9. Transcriptomic Analysis of Winged and Unwinged Sitobion Avenae Aphids

We further investigated the expression profiles of *S. avenae*, including different tissues (body, head, whole organism) of unwinged aphids and the whole organism of winged aphids. The expression profiles and differential expression analysis of unwinged aphids were conducted among three groups (body vs. head group, whole organism vs. body group and whole organism vs. head group). A total of 31,007 genes were identified, of which 5857, 5312, and 5778 genes had no expression among the body vs. head group, whole organism vs. body group, and whole organism vs. head group, respectively (Appendix A). Differential expression analysis found that there were 2865, 287, and 2114 differently expressed genes among the body vs. head group, whole organism vs. body group, and whole organism vs. head group, respectively (Figure 9A). The body vs. head group had the most differently expressed gene number. In detail, there were 1272 up-regulated expressed genes and 1593 down-regulated expressed genes in the body vs. head group, 252 were up-regulated expressed genes and 35 were down-regulated expressed genes in the whole organism vs. body group, and 1091 were up-regulated expressed genes and 1023 were down-regulated expressed genes in the whole organism vs. head group. We found that down-regulated expressed genes were larger than up-regulated expressed genes in the body vs. head group. However, up-regulated expressed genes were larger than down-regulated expressed genes in the whole organism vs. body and whole organism vs. head groups, indicating that up-regulated expressed genes were critical for whole organism tissue. Furthermore, these three groups had a unique gene set with 196 DEGs (Figure 9B). There were 1914 DEGs existing in both the body vs. head group and the whole organism vs. head group. There were 207 DEGs existing in both the whole organism vs. body group and the whole organism vs. head group. And there were 244 DEGs existing in both te body vs. head group and the whole organism vs. body group. The expression patterns of DEGs among different tissue groups were differential (Figure 9C–E). The intersection of DEGs among these three groups was used for the GO and KEGG enrichment analysis. Results found that DEGs were enriched in the go terms: lipase activity, passive transmembrane transporter activity, channel activity (molecular function), rhabdomere, extracellular space, obsolete extracellular region part (cellular component), phototransduction, detection of light stimulus, and rhodopsin-mediated signaling pathway (biological process) (Figure 9F). KEGG pathways, including sensory system, ion channels, amino acid metabolism, signaling, and cellular processes were significantly enriched (Figure 9G). Moreover, we found that there were 22, 2, and 11 differently expressed *SaveCYPs* in the body vs. head group, whole organism vs. body group, and whole organism vs. head group, respectively. These differently expressed *SaveCYPs* might play a significant role across the tissue development of *S. avenae* (Appendix A).

Additionally, the whole organism of unwinged and winged *S. avenae* aphids were used to analyze the expression profiles and differential expression analysis. A total of 31,007 genes were identified, of which 5380 genes had no expression (Appendix A). Differential expression analysis found that there were 233 differently expressed genes, of which 10 DEGs were up-regulated expressed genes and 223 DEGs were down-regulated expressed genes (Appendix A). It is obvious that down-regulated expressed genes were more prevalent than up-regulated expressed genes, indicating that these down-regulated DEGs might negatively regulate the wing growth and development of *S. avenae* (Appendix A). The results of GO and KEGG enrichment analysis found that these DEGs were enriched in the go terms: actin filament binding, transmembrane signaling receptor activity, structural molecule activity, signaling receptor activity (molecular function), external encapsulating structure, extracellular matrix, obsolete extracellular region part, extracellular region (cellular component), positive regulation of organelle assembly, regulation of actin filament polymerization, actin filament polymerization, regulation of actin polymerization or depolymerization, and regulation of actin filament length (biological process). KEGG pathways, including cardiac muscle contraction, exosome, enzymes with EC numbers, and lipid metabolism were significantly enriched (Appendix A). Moreover, we found that there was only one differently expressed gene (*SaveCYP39*) which was differently up-regulated expressed in the unwinged whole organism vs. winged whole organism group, indicating that *SaveCYP39* played a crucial regulatory role in the wing development of *S. avenae* (Appendix A).

### 2.10. Transcriptomic Analysis of Sitobion Avenae Aphids Fed on Wheat Treated with Imidacloprid

In order to investigate the metabolic detoxification function of *CYP450* genes in aphids towards insecticides, the RNA-seq analysis of *S. avenae* aphids fed on wheat treated with imidacloprid was conducted. Results showed that a total of 31,007 genes were identified, of which 6213 genes had no expression among imidacloprid treatment vs. control (Appendix A). Differential expression analysis found that there were 119 differently expressed genes, of which 60 DEGs were up-regulated expressed genes and 59 DEGs were down-regulated expressed genes (Figure 10A,B). The expression patterns of DEGs when *S. avenae* aphids fed on wheat treated with imidacloprid are shown (Figure 10C). The results of GO enrichment analysis found that these DEGs were enriched in the go terms: unfolded protein binding, DNA-binding transcription factor activity (RNA polymerase II-specific), DNA-binding transcription factor activity (molecular function), extracellular space, extracellular region (cellular component), protein folding, response to starvation, response to extracellular stimulus, multicellular organism aging, obsolete cofactor metabolic process, and response to external stimulus (biological process) (Figure 10D). KEGG pathways, including chaperones and folding catalysts, antigen processing and presentation, protein processing in endoplasmic reticulum, and protein families’ metabolism were significantly enriched (Figure 10E). Moreover, we found that there was only one differently expressed gene *SaveCYP48* (*g21172.t1)* was differently up-regulated expressed *S. avenae* aphids fed on wheat treated with imidacloprid, indicating that *SaveCYP48* played a crucial regulatory role when *S. avenae* aphids resisted the toxicity to imidacloprid (Appendix A).

## 3. Discussion

Cytochrome P450 monooxygenases (P450s) are a large gene superfamily of heme thiolate proteins in prokaryotes or eukaryotes [36]. In addition to detoxifying exogenous toxic compounds like insecticides and plant secondary insecticides, cytochrome P450s play a role in the synthesis and degradation of endogenous compounds in insects, including hormones and sex pheromones. They also mediate insect growth and development, host plant adaptation, and insecticide resistance [37]. To date, the *CYP450* gene family has been identified and characterized in prokaryote (cyanobacteria) [38] and fungi [39]. And the *CYP450* gene family has also been analyzed in various plants species, such as rice [40], tomato [41], wheat [42], maize [42], sorghum [43], soybean [44], tea [45], and others. In insects, it was reported that the *CYP450* gene family of *Drosophila melanogaster* [46], silkworm (*Bombyx mori*) [47], *Plutella xylostella* [48], *Scopula subpunctaria* [49], cotton leafhopper (*Amrasca biguttula*) [50], bees [51], and *Bemisia tabaci* [37] have been identified and analyzed. However, several aphid species had not been reported yet. In this study, we identified a total of 1100 *CYP450* genes in 19 aphid species at the genome level. The *CYP450* gene family of *A. pisum*, *M. persicae*, *A. Gossypii*, and *A. Glycines* owned 83,115,49 genes, and 68 members have been reported [52], while 66, 60, 54, and 60 *CYP450s* were identified in our results, which may be due to the strict parameters for identifying gene families. Furthermore, the 300 *CYP450* genes belonged six cereal crop aphids, including *S. avenae*, *S. graminam*, *S. miscanthi*, *R. padi*, *R. maidis,* and *D. noxia,* which we focused on. It is generally believed that the insect *CYP450* gene family consists of four branch clusters, namely the CYP2 clan, CYP3 clan, CYP4 clan, and the Mito clan [20]. According to the phylogenetic relationships and conserved motifs among six cereal crop aphids, these 300 *CYP450* genes were classified into four groups, which we named CYP I, CYP II, CYP III, and CYP IV. CYP I belonged to the CYP3 clan, CYP II belonged to the CYP4 clan, CYP III belonged to the Mito clan, and CYP IV belonged to the CYP2 clan. The classification was consistent with that of the above reported insects.

In insects, cytochrome P450s have a role in their growth and development. Previous research revealed that, in comparison to other developmental phases, the expression of the *TcCYP6BQ8* gene was comparatively higher in the early and late-larval stages of *Tribolium castaneum* [53]. In our study, we analyzed the expression profiles of *R. padi* from the mature embryo stage to the first instar stage, and found that 15 *RpadCYPs* were highly expressed at the mature embryo stage, suggesting that *CYP450* genes mainly participated in the growth and development of mature embryos. According to tissue expression profiles of *Tribolium castaneum*, *TcCYP6BQ8* is mostly expressed in the head and integument of both larvae and adults [53]. In another research, *Pocyp4d2* had the highest expression in the midgut of *Phortica okadai* [54]. It was reported that *SsubCYP341A* and *SsubCYP341B_ortholog1* were more highly expressed in the pheromone gland than in the female body of *Scopula subpunctaria* [49]. In our study, we obtained and analyzed the expression profiles of the unwinged *S. avenae* aphids among different tissues (body, head, and the whole organism). And we found that most of the *SaveCYPs* group had a higher expression level in the head than in the body, indicating that the *CYP450* genes preferred to be expressed in the head tissue of *S. avenae*. In addition, we found that *SaveCYP39* was down-regulated in the winged *Sitobion avenae* aphids, and thus *SaveCYP39* might negatively regulate the wing growth and development of *S. avenae*.

The primary detoxifying enzyme system in insects is cytochrome P450, which is involved in the metabolism of a variety of pesticides, as well as other exogenous and endogenous substances [21]. Increasing CYP450 enzyme activity and overexpressing *CYP450* genes were linked to pesticide resistance in a variety of insects [55]. Previous research revealed that the diamondback moth population resistant to chloramphenicol expressed much more of the P450 gene *CYP6BG1* than susceptible populations [56]. When *Tribolium castaneum* was exposed to turpentene-4-ol, the expression of *TcCYP6BQ8* was significantly induced. And silencing the cytochrome P450 gene *TcCYP6BQ8* increased the larval mortality rate induced by turpentene-4-ol from 47.78% to 66.67% [53]. In addition, cotton specific aphids and cucumber specific aphids fed on epigallocatechin gallate (EGCG) and cucurbitacin B (CucB) significantly induced the expression of *AgoCYP6CY19* [57]. When fourth-instar larvae were subjected to LC_50_ dosage levels of indoxacarb, the death rate rose due to RNA interference-mediated *CYP6AE68* silencing [55]. In our study, we analyzed transcriptome of *D. noxia* and *S. graminum* fed on wheat plants containing different resistance genes, *Dn4* or *Dn7,* compared to *Dn0*. *DnoxCYP19* (*Dnox005792.1*) was differently up-regulated expressed when *D. noxia* fed on wheat plants containing different resistance genes, *Dn7*, indicating that *DnoxCYP19* might participate in weakening the resistance gene *Dn7*. However, *SgrmCYP16* (*Sgrm006932.1*) was differently up-regulated expressed when *S. graminum* fed on wheat plants containing different resistance genes, *Dn4*, indicating that *DnoxCYP19* might participate in weakening the resistance gene, *Dn4*. To explore whether the *CYP450* gene family was related to the detoxification of insecticides in aphids, we also analyzed the transcriptome of *S. avenae* aphids fed on wheat treated with imidacloprid. Finally, we found that *SaveCYP48* (*g21172.t1)* was differently up-regulated expressed in *S. avenae* aphids fed on wheat treated with imidacloprid, indicating that *SaveCYP48* might play a crucial regulatory role when *S. avenae* aphids resisted the toxicity to imidacloprid. In conclusion, our study systematically identified and analyzed *CYP450* genes in 19 aphid species, providing some useful clues for the further molecular and functional identification of the *CYP450* gene family. The transcriptome profiles of aphids at different developmental stages, different tissues, and when fed on wheat plant containing resistance genes or imidacloprid provide new potential targets for aphid control and management.

## 4. Materials and Methods

### 4.1. Genome-Wide Identification of CYP450 Genes in 19 Aphid Species

The genome, CDS, protein, and genome annotation file of 18 aphid species, except *S. avenae,* were obtained from the InsectBase 2.0 (http://v2.insect-genome.com/ (accessed on 15 November 2022)) as the local database. The genome, CDS, protein, and genome annotation file of *S. avenae* were obtained from the figshare website (https://figshare.com/ (accessed on 13 November 2022)) [58] (Appendix A). Then, using the HMMER 3.0 tool with e-value < 1 × 10^−10^, the CYP450 typical domain (PF00067) was retrieved from the PFAM database (http://pfam-legacy.xfam.org/ (accessed on 16 November 2022)) and used as the query to search against the protein database of every species of aphid. The protein sequences identified with above method were regarded as candidates of *CYP450* gene family. To confirm the existence of the Cytochrome P450 superfamily domain, the candidates were then submitted to the NCBI-CDD (https://www.ncbi.nlm.nih.gov/cdd/ (accessed on 2 December 2022)) and InterPro websites (http://www.ebi.ac.uk/interpro/ (accessed on 2 December 2022)). The proteins harboring the complete Cytochrome P450 superfamily domain were considered to be *CYP450* genes. The putative CYP450 proteins were uploaded to the ExPASy database (https://web.expasy.org/protparam/ (accessed on 2 December 2022)) in order to calculate the grand average of hydropathicity (GRAVY), molecular weight (MW), and theoretical isoelectric point (pI). The WoLF PSORT program was used to predict the subcellular localization of them.

### 4.2. Phylogenetic Relationship, Gene Structure and Conserved Motif Analysis of CYP450 Genes among Six Cereal Crop Aphid Species

To understand the evolutionary relationship of *CYP450* genes among cereal crop aphids, including *D. noxia*, *R. maidis*, *R. padi*, *S. graminum*, *S. miscanthi*, and *S. avenae*, multiple sequence alignments of the discovered CPY450 proteins of these cereal crop aphids were carried out using the ClustalW v2.0 program [59]. The phylogenetic tree was created using the MEGA-X program using the neighbor-joining approach with a 1000 replication bootstrap [60]. The tree of phylogenetic relationships was improved using EVOLVIEW (https://evolgenius.info//evolview-v2/#login (accessed on 7 December 2022)). The chromosomal location and exon-intron structures of *CYP450* genes were obtained from genome annotation files (gff3). MapGene2Chromosome v2.0 (http://mg2c.iask.in/mg2c_v2.0/ (accessed on 16 January 2023)) was used to display the physical positions of the *CYP450* genes on the chromosomes, and Gene Structure Display Server (GSDS2.0) (https://gsds.gao-lab.org/ (accessed on 8 December 2022)) was used to display the exon-intron structures. The MEME v5.2.0 program was used to find the conserved motifs of CYP450 proteins, with a maximum of ten motifs allowed [61].

### 4.3. Gene Duplication and Synteny Analysis of CYP450 Genes among Six Cereal Crop Aphid Species

To understand expansion events of the *CYP450* gene family, including gene duplication and tandem duplication, MCScanX software was utilized to evaluate syntenic relationships among all *CYP450* genes present in cereal crop aphids. The Circos tool was used to display the linked gene pairs. The software MCScan (Python-jcvi) was utilized to identify syntenic links between the various genomes of cereal crop aphids. Using the KaKs_Calclator 2.0 program, the synonymous (Ks) and nonsynonymous (Ka) substitution rates were determined.

### 4.4. RNA-seq Data Analysis of Four Cereal Crop Aphid Species

To get additional insight into the expression patterns and spatial-temporal expression of these *CYP450* genes among cereal crop aphids, we collected RNA-seq data of four cereal crop aphids, including *R. padi*, *D. noxia*, *S. graminum,* and *S. avenae*, which were obtained from Sequence Read Archive database (http://www.ncbi.nlm.nih.gov/sra (accessed on 9 February 2023)) (Appendix A). The wheat plants harboring distinct resistance genes *Dn0*, *Dn4*, or *Dn7* were fed to the aphid biotypes of *S. graminum* and *D. noxia*, identified as *S. graminum* biotype I and *D. noxia* biotype 1, respectively. The transcriptomes of *S. graminum* biotype I and *D. noxia* biotype 1 were compared using the wheat varieties Yuma, which contains no resistance genes (*Dn0*), Yumar, which contains a *Dn4* resistance gene [62], and 94M370, which contains a *Dn7* gene for resistance to *D. noxia* biotype 2 [63].

The RNA data of *R. padi* were at the stages of the mature embryo and first instar. And the RNA data of *S. avenae* contained the body, head, and whole organism of unwinged and winged aphids. Moreover, we collected the RNA-seq data of *S. avenae* aphids fed with wheat treated with imidacloprid. Hisat2 was used to map this RNA-seq data to the appropriate local reference genome. StringTie v2.1.2 was used to determine the FPKM of each gene (fragments per kilobase of transcript per million fragments of mapped reads) value [64]. The featureCounts program was used to compute raw counts.

### 4.5. Differential Expression Analysis among Four Cereal Crop Aphids

DESeq was used to filter the differentially expressed genes (DEGs) among the several comparison groups according to the following standards: |log2 FC| ≥ 1, and *p* value < 0.05. Using the Z-score approach, the FPKM of all DEGs across various comparison groups were normalized. The pheatmap package and ggplot2 package in R software (4.0.3) were used to create a heatmap and a volcano plot to visualize the expression patterns.

### 4.6. GO Enrichment and KEGG Pathway Enrichment of DEGs among Four Cereal Crop Aphids

The DEGs were subjected to GO enrichment and KEGG pathway enrichment analysis by utilizing the TBtools program [65]. A BackGround file (Query2Go and Query2Knum) was extracted from the gene annotation file. The bubble charts of GO enrichment and KEGG enrichment were visualized using the clusterProfiler package.

### 4.7. Expression Patterns of Differential Expressed CYP450 Genes among Four Cereal Crop Aphids

In order to comprehend the patterns of spatial-temporal expression of *CYP450* genes among cereal crop aphids, the differential expressed *CYP450* genes between the above different comparison groups were extracted to further display the expression profiles of *CYP450* family genes. Through the Z-score approach, the FPKM of CYP450 family genes was normalized across several comparison groups. The pheatmap package and ggplot2 package in R software (4.0.3) were used to create a heatmap, and a volcano plot was used to visualize the expression patterns.

## 5. Conclusions

This is the first study to identify the cytochrome P450 gene family in 19 aphid species at the genome level. A total of 1100 *CYP450* genes were identified in 19 aphid species, and 300 *CYP450* genes were belonged to 6 cereal crop aphid species. The members of *S. flava* had the largest average value of the gene length, CDS length, protein size, and molecular weight. However, the members of *A. solani* had the lowest average value of the gene length, CDS length, protein size, and molecular weight. Subcellular localization predictions revealed that most of CYP450 proteins were located in the endoplasmic reticulum, indicating that they mainly played a role in the endoplasmic reticulum. The predicted GRAVY value of all these CYP450 proteins were minus, indicating they attract water molecules or easily dissolve in water. According to phylogenetic relationship, these 300 *CYP450* genes were further divided into four subfamilies (CYPI, CYPII, CYPIII, and CYPIV). The genomic organizations, exon-intron structures, and conserved motifs of the same subfamilies were similar. Almost all members of CYP I subfamily had all of the 10 motifs, the members of CYP II subfamily lacked motif 5 and motif 8, the members of CYP III subfamily lacked motif 5, and the members of CYP IV subfamily lacked motif 4 and motif 7, indicating that there are different motif organizations among these four subfamilies. The *CYP450* members of *R. maidis*, *R. padi,* and *S. graminum* always clustered together on the same clade, indicating that *R. maidis*, *R. padi,* and *S. graminum* had a closer genetic relationship. And the *CYP450* members of *D. noxia*, *S. miscanthi,* and the *S. avenae* always clustered together on the same clade, indicating that *D. noxia*, *S. miscanthi,* and *S. avenae* had a closer genetic relationship. The chromosome localizations and distribution of *CYP450* genes among six cereal crop aphid species were differential. The gene expansion events of *CYP450s* in *R. maidis* and *S. avenae* possessed both gene duplications and tandem duplications, while only tandem duplications occurred in *D. noxia*, *R. padi*, *S. graminum,* and *S. miscanthi*. Moreover, three continuous colinear gene pairs among six cereal crop aphid species were found, which were very conserved among these species. In addition, most of duplicated *CYP450* genes pairs across the whole genome of *S. avenae*, *R. maidis*, *D. noxia*, *R. padi*, *S. graminum,* and *S.miscanthi* underwent purifying selection during evolution. Furthermore, we systematically investigated the expression profiles of four cereal crop aphids, including *R. padi*, *D. noxia*, *S. graminum,* and *S. avenae*. Differential expression analysis provided growth stage specificity genes, tissue specificity genes, organ specificity genes, and some detoxification metabolic genes among these four cereal crop aphids. The differentially expressed *CYP450* genes from all the DEGs were obtained, which provided some useful clues for further study to the functions of the *CYP450* gene family, and provided some potential candidates for aphid management and control.

## Figures and Tables

**Figure 1 ijms-25-06668-f001:**
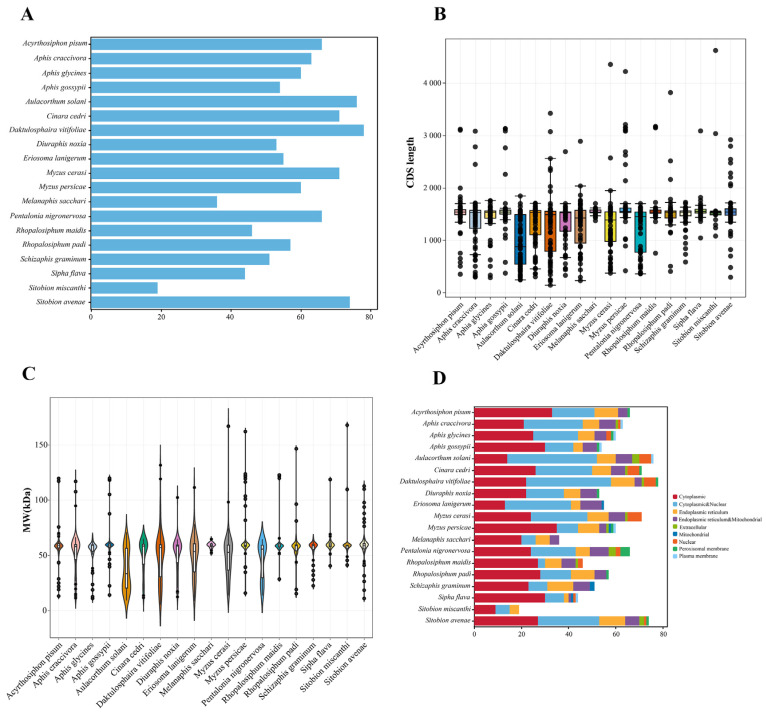
Basic information of Cytochrome P450 superfamily (*CYP 450*) in 19 aphid species. (**A**) The number of all *CYP450* genes in 19 aphid species. (**B**) The CDS length of all *CYP450* genes in 19 aphid species. (**C**) The molecular weight of all CYP450 proteins in 19 aphid species. (**D**) Subcellular localization of all CYP450 proteins in 19 aphid species.

**Figure 2 ijms-25-06668-f002:**
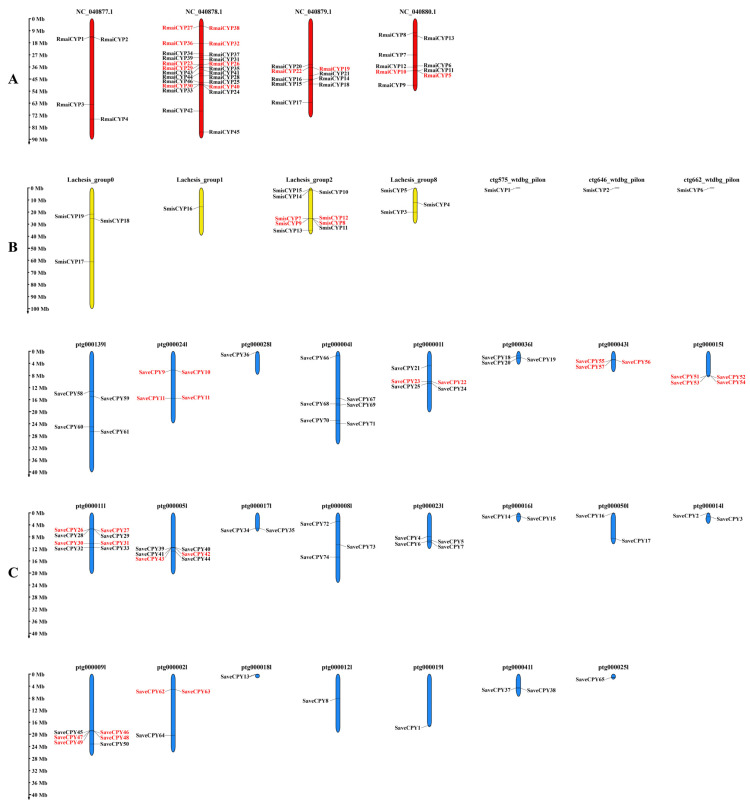
Distribution of the identified *CYP450* genes across cereal crop aphid genomes of (**A**) *R. maidis*, (**B**) *S. miscanthi,* and (**C**) *S. avenae*. All aphid scaffolds are drawn to scale according to their actual physical lengths. The gene pairs marked in red are tandem duplications.

**Figure 3 ijms-25-06668-f003:**
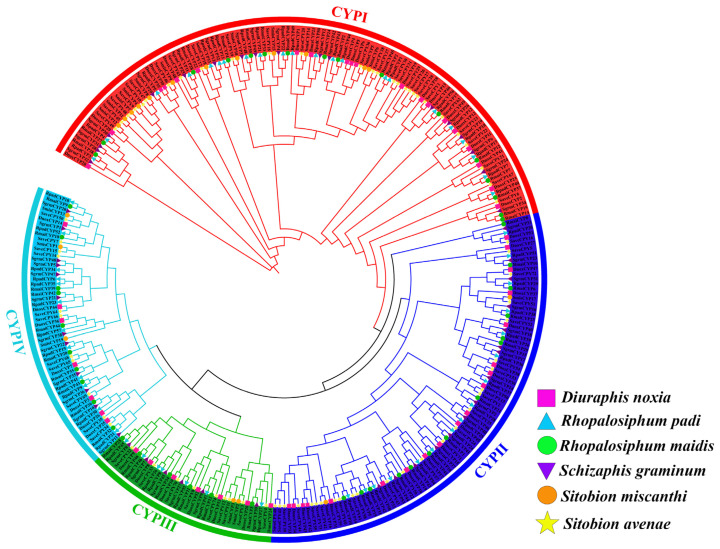
Phylogenetic analysis of *CYP450* gene family in cereal crop aphid species. The magnitude of genetic change is represented by the branch length. The subfamilies CYP I, CYP II, CYP III, and CYP IV are represented by the red, purple, green, and blue clades, respectively.

**Figure 4 ijms-25-06668-f004:**
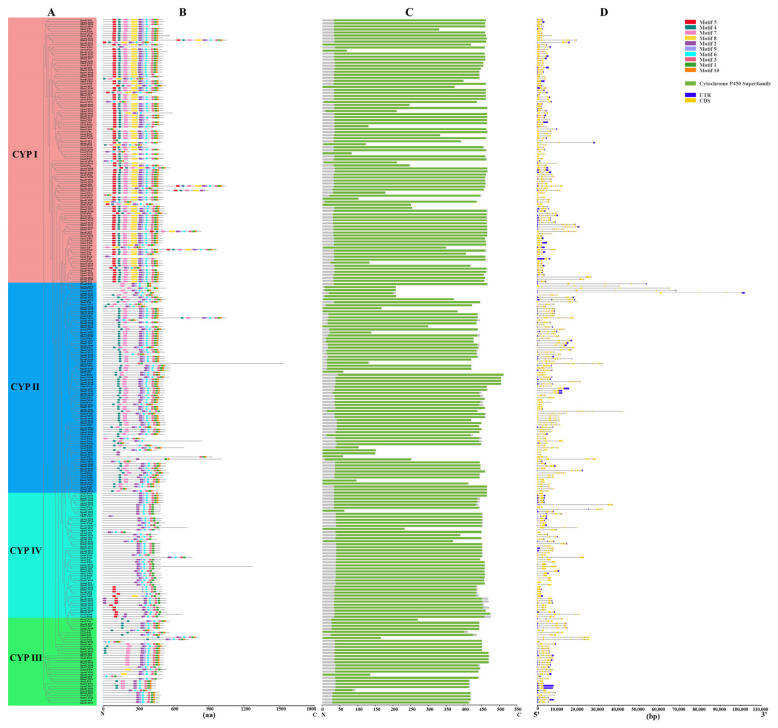
Phylogenetic relationship (**A**), conserved motifs (**B**), conserved domain (**C**), and gene structure (**D**) of *CYP450* genes in cereal crop aphid species. Ten identified motifs were represented by different colors. The UTRs and CDSs are shown by the blue and yellow rectangles, respectively. Introns are represented by the grey lines. aa: amino acid, bp: base pairs, N: N-terminus of amino acids, C: C-terminus of amino acids.

**Figure 5 ijms-25-06668-f005:**
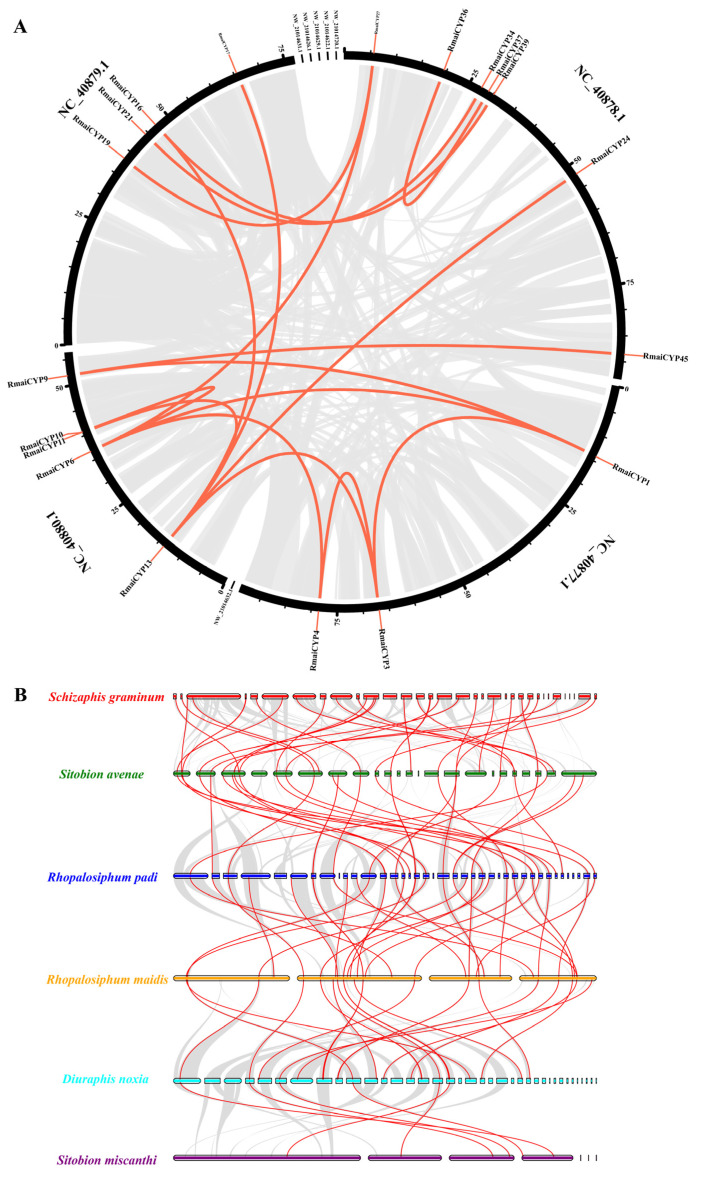
Duplication events of *CYP450* genes in the genome of *R. maidis,* and synteny analysis of *CYP450* genes among the different genomes of six cereal crop aphid species. (**A**) Black circos represent each scaffold of *R. maidis*. Duplicate pairs are shown with red lines. (**B**) The collinear blocks of the whole genome across six cereal crop aphid species are shown by the gray lines in the background. Collinear couples in several species are represented by red lines. The scaffolds of various cereal crop aphid species are represented by different colored bars.

**Figure 6 ijms-25-06668-f006:**
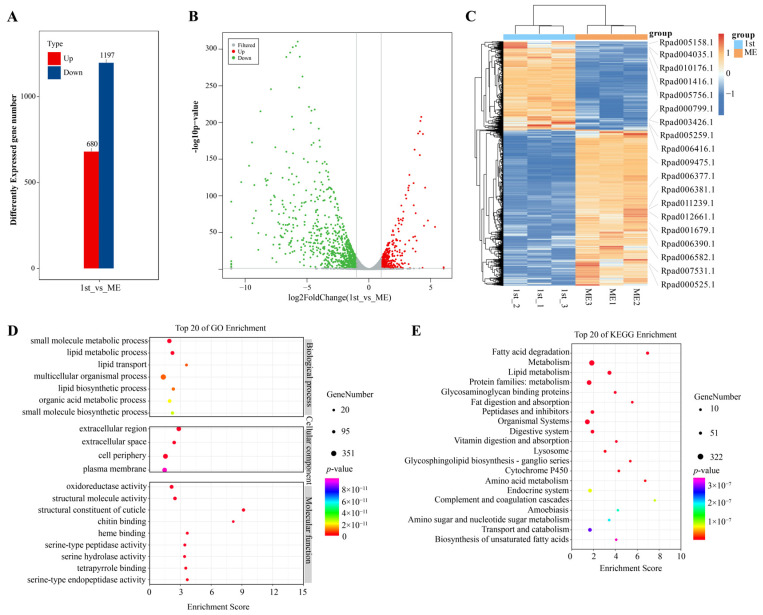
Transcriptomic overview of *R. padi* at different developmental stages. (**A**) Total number of significantly up- or down-regulated genes. (**B**) Volcano plots of DEGs between the first instar and mature embryo. (**C**) Expression patterns of *R. padi* DEGs at different developmental stages. Log2 expression values are represented using the color scale. The expression level transforms log2 values, and is equivalent to the mean values. The 1st label represents the first instar of *R. padi.* ME represents mature embryo of *R. padi.* (**D**) GO enrichment analysis of the DEGs between the first instar and the mature embryo. (**E**) KEGG enrichment analysis of the DEGs between the first instar and the mature embryo.

**Figure 7 ijms-25-06668-f007:**
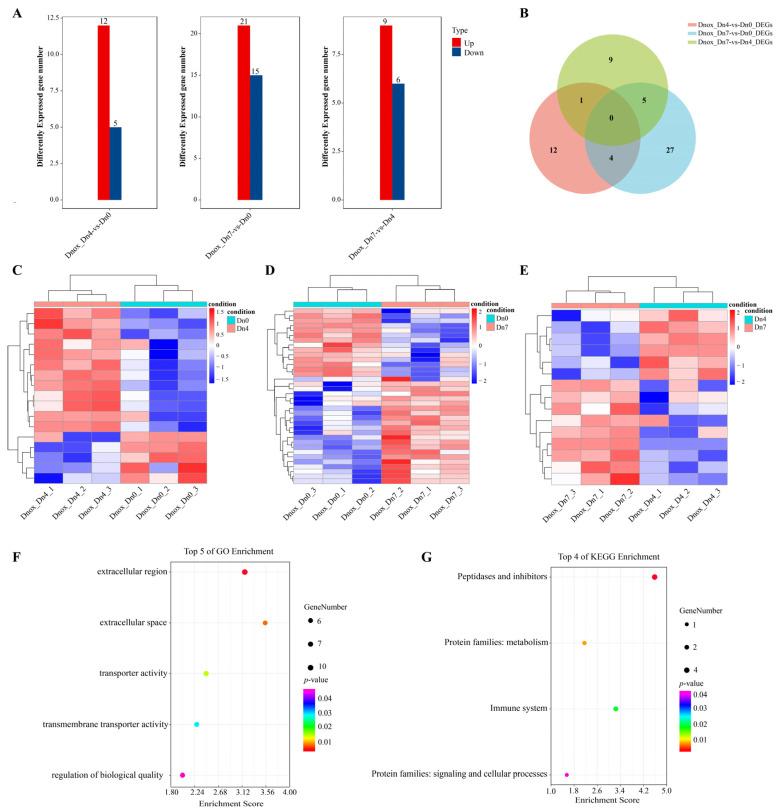
Transcriptomic overview of *D. noxia* fed on wheat plants containing different resistance genes *Dn0*, *Dn4,* or *Dn7*. (**A**) Total number of significantly up- or down-regulated genes between different groups. (**B**) Venn diagram of DEGs between different groups. (**C**–**E**) Expression patterns of *D. noxia* DEGs between different groups. Log2 expression values are represented using the color scale. The expression level transforms log2 values, and is equivalent to the mean values. (**F**) GO enrichment analysis of the union DEGs among all the groups. (**G**) KEGG enrichment analysis of the union DEGs among all the groups.

**Figure 8 ijms-25-06668-f008:**
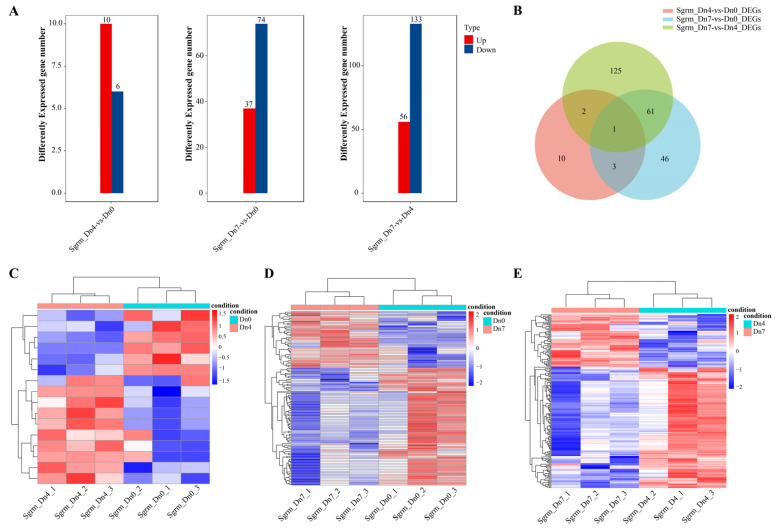
Transcriptomic overview of *S. graminum* fed on wheat plants containing different resistance genes *Dn0*, *Dn4,* or *Dn7*. (**A**) Total number of significantly up- or down-regulated genes between the different groups. (**B**) Venn diagram of the DEGs between different groups. (**C**–**E**) Expression patterns of *S. graminum* DEGs between different groups. Log2 expression values are represented using the color scale. The expression level transforms log2 values, and is equivalent to the mean values.

**Figure 9 ijms-25-06668-f009:**
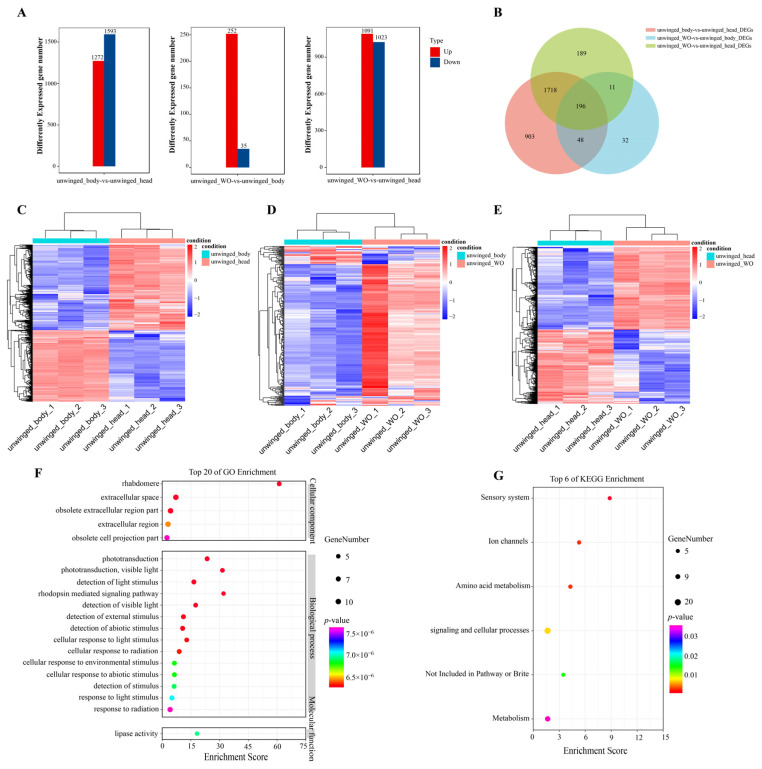
Transcriptomic overview of unwinged *S. avenae* aphids. (**A**) Total number of significantly up- or down-regulated genes among different aphid tissue groups. (**B**) Venn diagram of DEGs between different tissue groups. (**C**–**E**) Expression patterns of the DEGs of unwinged *S. avenae* aphids between different tissue groups. Log2 expression values are represented using the color scale. The expression level transforms log2 values, and is equivalent to the mean values. (**F**) GO enrichment analysis of the intersection DEGs among all the groups. (**G**) KEGG enrichment analysis of the intersection DEGs among all the groups.

**Figure 10 ijms-25-06668-f010:**
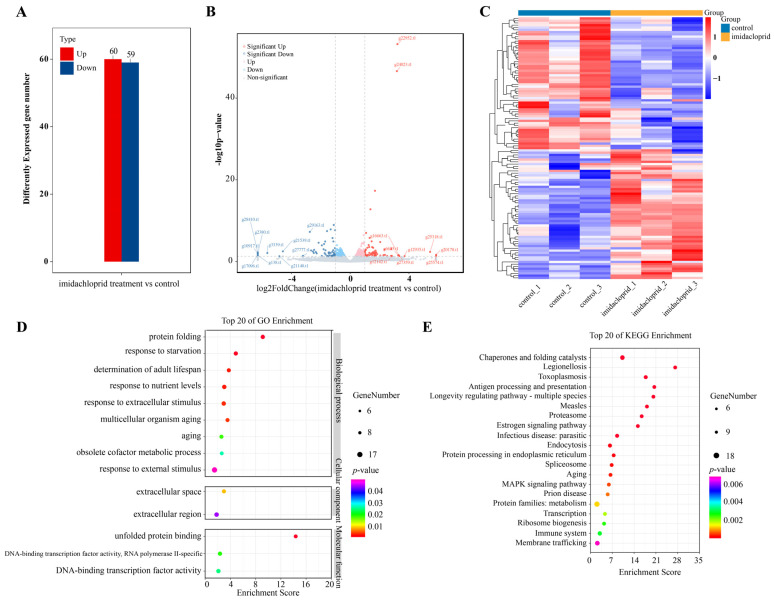
Transcriptomic overview of *S. avenae* aphids fed on wheat treated using imidacloprid. (**A**) Total number of genes that were significantly up- or down-regulated. (**B**) Volcano plots of differentially expressed genes (DEGs) between imidacloprid treatment and control. (**C**) Expression patterns of *S. avenae* aphids fed on wheat treated using imidacloprid. The color scale represents Log2 expression values. (**D**) GO enrichment analysis of the DEGs between imidacloprid treatment and control. (**E**) KEGG enrichment analysis of the DEGs between imidacloprid treatment and control.

## Data Availability

All data supporting the findings of this study are available in the paper and its Appendix A. Further information may be obtained from the corresponding author, Wanquan Ji.

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
