# Peer review of "Genome-Wide Comparative Analysis of the Cytochrome P450 Monooxygenase Family in 19 Aphid Species and Their Expression Analysis in 4 Cereal Crop Aphids"

_ijms, 2024, doi:10.3390/ijms25126668_

Round 1

Reviewer 1 Report

Comments and Suggestions for Authors

This paper provides a comprehensive summary of aphid P450 genes, surveying the genomes of 19 aphid species, and following up with some detailed analysis of expression across developmental stages, tissues and genetic backgrounds of resistance.

Although the analysis appears broadly good, and the data presented will be useful to researchers interested in crop pests and insecticide resistance, the quality of writing and the logic of how results are presented need to be significantly improved.

Points to be addressed:

1) The introduction lists a lot of aphid species and the harm they cause to crops (more than a page is devoted to this). This provides too much detail about justification for the research before we get into what is actually being done. It would perhaps work better to frame this by summarising the different ways aphid pests can damage crops (viral transmission, diminishing yield, etc...) and to list species that have these effects alongside.

In contrast, very little is said about current understanding of P450s - and again what is said reads as a list of examples. The introduction would benefit from including a lot of the first paragraph of the discussion - lots of questions raised when reading the results are answered at the start of the discussion, and it would help to have this context earlier in the paper. How do P450s act? What do we already know about P450s in aphids (for example I know they have been surveyed in A. pisum, see Zhang et al 2010 and Duvaux et al 2015)? What do we know about P450s in insects more broadly (e.g. 4 families)? And why is it useful to have this survey of P450s in aphids, what will it allow us to do?

2) The abstract and conclusion paragraphs are very similar.

3) Results are overly wordy - a lot of what is said could be summarised in tables or figures (and is), so probably doesn't need repeating in text too.

4) It wasn't clear to me from the methods whether the authors conducted the RNAseq experiments themselves. If they did, there needs to be more detail about experimental design, methods of RNA extraction etc..., if they did not, then there needs to be clear citation of the papers where these details can be found.

5) The differential expression analyses were interesting, but appart from description of the P450 genes they found they were perhaps not the most relevant to the rest of the paper. I would advise a closer focus on the expression of the P450s - is there evidence for enrichment in any of the comparisons, for example?

6) It would be helpful to know how representative these 19 species are of aphids as a whole.

7)The phylogeny is interesting, but I have a couple of questions. Firstly, how do the groups identified here relate to the 4 main clades of CYP450s seen in insects? Secondly, I wonder if the tree would work better rooted on an outgroup. Currently the CYPI and CYPII groups are both paraphyletic, might an outgroup help with this?

8) You mention that analysis of P450s has not been done before in aphids, but it has been done for some species. It would be useful to cite papers where P450s have been summarised in other aphid species, and to compare your results to those - e.g. did you ID the same number of P450s in A. pisum as Zhang did? This would (hopefully) give support to your methods of gene identification.

Comments on the Quality of English Language

The English Language needs some work, sometimes just basic grammar (e.g. there are many places where articles are missing) or spelling (e.g. "species" not "specie"), but also occasionally meaning is obscured by garbled grammar, or sentences are redundant (i.e. saying the same thing in two consecutive sentences). Getting some help from a speaker of English as a first language would be a good idea.

Just a few examples here, not an exhaustive list:

Line 14: “classification and expression profiles of CYP450 gene family members (or the CYP450 gene family) in aphid species”

Line 15: “this is the first study”

Line 17: “300 CYP450 genes belonged” --> not "were belonged"

Line 28-29: "“…their expression patterns were showed related functions and pathways of the CYP450s were revealed by GO and KEGG….” —> confusing

Line 32 “Moreover, these differentially expressed ……” duplicates previous sentence for meaning - not needed.

Line 38: “The most significant pest in global agriculture —> this is a big claim! One of the most significant.

Line 39: “invasion of phloem sap” —> invasion is not the right word here.

Line 657: "The tree of phylogenetic relationship was beautified" --> unusual phrasing.

Line 538: “From prokaryotes to eukaryotes, a vast and complex superfamily of creatures is comprised of cytochrome P450s”
—> Confusing, do you mean P450s are present in organisms across the tree of life?

Author Response

Dear reviewers:

Thank you for arranging a timely review for our manuscript. And we are very glad to receive your reply again. On behalf of my co-authors, we thank you very much for giving us an opportunity to revise our manuscript. We are very grateful for your comments regarding our manuscript entitled “Genome-wide comparative analysis of the Cytochrome P450 monooxygenase family in 19 aphid species and their expression analysis in 4 cereal crop aphids” (Manuscript ID: ijms-3027395). All your suggestions are all valuable and very helpful for futher revising and improving our paper. We have studied comments carefully and have made corrections which we hope meet with your approval. We have revised it on the basis of the manuscript for revision in system. The revised portions are marked in red in the paper. A point by point response to the editorial and reviewers' comments are as follows:

1) The introduction lists a lot of aphid species and the harm they cause to crops (more than a page is devoted to this). This provides too much detail about justification for the research before we get into what is actually being done. It would perhaps work better to frame this by summarising the different ways aphid pests can damage crops (viral transmission, diminishing yield, etc...) and to list species that have these effects alongside.

Response: Thank you for your professional and constructive suggestions in the improvements of my introduction. We have rewritten the introduction section based on your suggestions in Line 37-136 from Line 38-139 in the last version. The first paragraph is about the harm of aphids, the second paragraph is an overview of the cytochrome P450, the third paragraph is about the research progress of the cytochrome P450, and the fourth paragraph is the purpose and significance of this study.

2) The abstract and conclusion paragraphs are very similar.

Response: Thank you for your significant reminding. We have rewritten the conclusion paragraphs in Line 972-1011.

3) Results are overly wordy - a lot of what is said could be summarised in tables or figures (and is), so probably doesn't need repeating in text too.

Response: Thank you for your valuable advice. We have deleted the wordy phrases and sentences in the Results section.

4) It wasn't clear to me from the methods whether the authors conducted the RNAseq experiments themselves. If they did, there needs to be more detail about experimental design, methods of RNA extraction etc..., if they did not, then there needs to be clear citation of the papers where these details can be found.

Response: Thank you for pointing this out. In this study, we did not conduct RNA seq experiments. Instead, we downloaded publicly available transcriptome data that has been published or released in public database.

5) The differential expression analyses were interesting, but appart from description of the P450 genes they found they were perhaps not the most relevant to the rest of the paper. I would advise a closer focus on the expression of the P450s - is there evidence for enrichment in any of the comparisons, for example?

Response: Thanks for your valuable counsel. Our initial idea and purpose were also to study the expression of CYP450 genes and explore differentially expressed genes. The expression profiling analysis and differential expression analysis is to further investigate the expression of the P450s and differentially expressed CYP450 genes.The results of the expression of the P450s and differentially expressed CYP450 genes showed in Figure S9, Figure S11 and Figure S13.

6) It would be helpful to know how representative these 19 species are of aphids as a whole.

Response: Thank you for your encouragement and support.

7)The phylogeny is interesting, but I have a couple of questions. Firstly, how do the groups identified here relate to the 4 main clades of CYP450s seen in insects? Secondly, I wonder if the tree would work better rooted on an outgroup. Currently the CYPI and CYPII groups are both paraphyletic, might an outgroup help with this?

Response: It is generally believed that the insect P450s gene family consists of four branching clusters, namely CYP2 clan, CYP3 clan, CYP4 clan, and Mito clan. In our study, CYPⅠbelonged to CYP3 clan, CYPⅡ belonged to CYP4 clan, CYP Ⅲ belonged to Mito clan and CYPⅣ belonged to CYP2 clan,which we hava added to the Dissussion section in 798-800. We agree with you that the tree would work better rooted on an outgroup, but we are not very familiar with using outliers for analysis, and we do not know which sequence to choose as the outgroup. In this study, we combined the results of phylogenetic tree and motif identification to distinguish CYP I and CYP Ⅱ. CYPⅠgroup owned motif5 and motif8, but CYPⅡdid not own these two motifs.

8) You mention that analysis of P450s has not been done before in aphids, but it has been done for some species. It would be useful to cite papers where P450s have been summarised in other aphid species, and to compare your results to those - e.g. did you ID the same number of P450s in A. pisum as Zhang did? This would (hopefully) give support to your methods of gene identification.

Response: Thank you for your valuable advice. We are vrey sorry that we did not retrieve the CYP450 gene family of aphids before. We have summarised the CYP450 genes of other aphid species and compared to our results in the Dissussion section in Line 761-764.

Comments on the Quality of English Language

The English Language needs some work, sometimes just basic grammar (e.g. there are many places where articles are missing) or spelling (e.g. "species" not "specie"), but also occasionally meaning is obscured by garbled grammar, or sentences are redundant (i.e. saying the same thing in two consecutive sentences). Getting some help from a speaker of English as a first language would be a good idea.

Response: Thank you for your suggestions on The English Language, which should undergo extensive English revisions. The language issues you pointed out have also been corrected. In addition, we invited a proficient English teacher and an international student from our laboratory to polish the language.

Line 14: “classification and expression profiles of CYP450 gene family members (or the CYP450 gene family) in aphid species”

Response: Thank you for pointing this out. The article is missing here. We have corrected this mistakes in Line 14.

Line 15: “this is the first study”

Response: Thank you for pointing this out. The article is missing here. We have corrected this mistakes in Line 15.

Line 17: “300 CYP450 genes belonged” --> not "were belonged"

Response: Thank you for pointing this out. We have corrected this mistakes in Line 17.

Line 28-29: "“…their expression patterns were showed related functions and pathways of the CYP450s were revealed by GO and KEGG….” —> confusing

Response: Thank you for pointing this out. We have deleted the confusing words“ were revealed”in Line 28-29.

Line 32 “Moreover, these differentially expressed ……” duplicates previous sentence for meaning - not needed.

Response: Thank you for pointing this out. We have deleted repetitive sentences “were revealed”in Line 30-31.

Line 38: “The most significant pest in global agriculture —> this is a big claim! One of the most significant.

Response: Thank you for pointing this out. We have corrected this inaccurate phrase in Line 37.

Line 39: “invasion of phloem sap” —> invasion is not the right word here.

Response: Thank you for pointing this out. We have corrected this inaccurate phrase as “through sucking phloem sap”in Line 38.

Line 657: "The tree of phylogenetic relationship was beautified" --> unusual phrasing.

Response: Thank you for pointing this out. We have corrected this inaccurate word  “ beautified” as “improved”in Line 917.

Line 538: “From prokaryotes to eukaryotes, a vast and complex superfamily of creatures is comprised of cytochrome P450s”

—> Confusing, do you mean P450s are present in organisms across the tree of life?

Response: Thank you for pointing this out. There is indeed some confusion here. What we want to express is that the CYP450 gene is widely present in prokaryotes or eukaryotes. We have rephased this sentence in Line 748-749.

Thank you again for your positive and constructive suggestions on our manuscript. Meanwhile, we have studied comments of reviewers carefully and have made corrections. And we have provided a point by point response to the reviewers' comments in reviewers' letters. We hope you will find our revised manuscript acceptable for publication. We look forward to your early reply.

With best wishes,

Yours sincerely,

Zhenyu Wang

Reviewer 2 Report

Comments and Suggestions for Authors

The gene family of cytochrome P450 play important roles in pesticide resistance, plant allelochemical detoxification, and hormone metabolism catalysis. This study amained to identify the cytochrome P450 gene family in 19 aphid species and conducted bioinformatics analysis of the gene family. The highlight of the research is the identification of differential expression P450 among four aphids.

Major concerns:

The author needs to revise the introduction. The author used two large paragraphs to describe the harm and economic importance of various aphids, and only used the last two paragraphs to introduce p450. Furthermore, the purpose and significance of this study were not clearly stated in the introduction. I think the introduction should focus on introducing the research progress of P450 and the scientific issues that this study aims to address.

Author Response

Dear reviewers:

Thank you for arranging a timely review for our manuscript. And we are very glad to receive your reply again. On behalf of my co-authors, we thank you very much for giving us an opportunity to revise our manuscript. We are very grateful for your comments regarding our manuscript entitled “Genome-wide comparative analysis of the Cytochrome P450 monooxygenase family in 19 aphid species and their expression analysis in 4 cereal crop aphids” (Manuscript ID: ijms-3027395). All your suggestions are all valuable and very helpful for futher revising and improving our paper. We have studied comments carefully and have made corrections which we hope meet with your approval. We have revised it on the basis of the manuscript for revision in system. The revised portions are marked in red in the paper. A point by point response to the editorial and reviewers' comments are as follows:

Major concerns:

The author needs to revise the introduction. The author used two large paragraphs to describe the harm and economic importance of various aphids, and only used the last two paragraphs to introduce p450. Furthermore, the purpose and significance of this study were not clearly stated in the introduction. I think the introduction should focus on introducing the research progress of P450 and the scientific issues that this study aims to address.

Response: Thank you for your professional and constructive suggestions in the improvements of my introduction. We have rewritten the introduction section based on your suggestions in Line 37-136 from Line 38-139 in the last version. The first paragraph is about the harm of aphids, the second paragraph is an overview of the cytochrome P450, the third paragraph is about the research progress of the cytochrome P450, and the fourth paragraph is the purpose and significance of this study.

Round 2

Reviewer 1 Report

Comments and Suggestions for Authors

Thanks for the changes you have made to this, it is very much improved.

I think the introduction reads much better now, and throughout you've done a good job of reducing wordy-ness.

A couple of specific points:

1) The results section still repeats in the written sections things which are shown well in figures and tables (e.g. you still give a very detailed description of numbers and size of CYP genes and proteins, lines 134 to 147, and 163-176 in particular). A lot of this is summarised well in figure 1, and doesn’t really need to take up all this space in the manuscript. I would advise removing a lot of the description and referring the reader to the relevant figures and tables instead.

2) The paragraph "Chromosome localizations of CYP450 genes among 6 cereal crop aphid species"

Distribution is interesting, but are these scaffolds equivalent to chromosomes? If not, then simply describing how many scaffolds contain CYP450s is not very useful. It would be better to see some estimation of clustering – do the genes show significant clustering?

In a similar vein, figure 2 shows identified CYP450 genes on the scaffolds where they were found. But I don’t think these are chromosomes, and so this image could be a bit misleading about how clustered or not the genes are, or what proportion of the genome they cover. Would it be possible to place the genes on chromosomes? E.g. in the R. maidis genome they have clustered contigs into 4 chromosomes & the S. miscanthi genome has 9 scaffolds (chromosomes?). Or if this is not possible, can you make it a bit more clear that you are only showing scaffolds containing CYP genes - e.g. in the figure, state how many scaffolds total each genome has, and give the proportion of scaffolds containing CYP genes.

3) Lines 455-457: interesting to see there were CYPs present in this gene set. Do you find more than expected by chance? You could test for this using an enrichment test, to match the GO and KEGG enrichment analyses.

I started to edit the spelling and grammar, and include some comments in the next section. However - this is just a selection of edits I identified, and I mostly focussed on the start. Someone will need to check later sections.

Comments on the Quality of English Language

Thank you for your work on this - it is definitely improved. However, there are still quite a lot of errors (especially in presence/absence of articles, and in singular vs plural). I have included points where I identified things to change (below) but this is not an exhaustive list, and I only did this for the abstract and introduction. Someone else will need to do the copy editing properly.

Abstract

Line 21: “the expansion of the CYP450 gene family was primarily caused…”

Numbers should be written as words for all numbers under nine – e.g. “three continuous colinear”, “four cereal crop aphids…”.

Line 28 & 29: still confusing, I don’t really understand what you mean with this sentence “Meanwhile, their expression patterns were showed related functions and pathways of the CYP450s by GO and KEGG enrichment analysis.”

Also line 29-30: “Above all, we picked the differentially expressed CYP450 genes from all the DEGs.” Does this mean that you found enrichment for CYP450 genes in the DEGs? Or just that you identified CYP450 genes which were differentially expressed?

Better to present results in the present tense:

Line 30: “these differentially expressed CYP450 genes provide some new…”.

Line 31: “this work establishes…

Introduction

Line 37: “significant pests in global agriculture”.

Line 40: “Pea aphids (Acyrthosiphon pisum) feed on legume…”.

Line 43: “Soybean aphids (), suck…”

For all the different aphid examples… Needs to be either “XXX aphids feed”, or “The XXX aphid feeds either of these grammatical formats is correct.

Line 69: “has led”…

Line 73: “metabolic enzymes include Glutathione” (don’t need “mainly”).

Line 89: thank you for including information on the four main insect clades. I would just edit slightly “the insect P450 gene family consists of four clades: the CYP2 clan, the CYP3 clan, the CYP4 clan and the Mito clan”.

Line 93: “The role of the P450 enzyme system…”.

Line 118: “research into insect P450s” or “into the insect P450 gene family”.

Line 121: “the genomic organization”

Results

A number of places where “specie” needs to be changes to “species”.

Line 133: “species with the lowest number of CYP450 genes”.

Line 153: “in the endoplasmic reticulum”.

Line 156: “All CYP450 proteins had a negative projected GRAVY value, which suggested…”.

Line 264: Unclear what this sentence intends to say. “Chromosome reduplication process”?

Line 309: Transcriptomic analysis of Rhopalosiphum padi at different developmental stages